# Embodied metaphor in communication about lived experiences of the COVID-19 pandemic in Wuhan, China

Yu Deng[1]*, Jixue Yang[2], Wan Wan[3]

**1** College of Language Intelligence, Sichuan International Studies University, Chongqing, China, **2** School of English Studies, Sichuan International Studies University, Chongqing, China, **3** College of Foreign Languages, Huaqiao University, Quanzhou, Fujian, China

* dengyu@sisu.edu.cn

**Data Availability Statement:** The minimal dataset is available within the paper. Additional data is available in figshare: https://doi.org/10.6084/m9.figshare.17152463.

## Abstract

The study investigated how a group of 27 Wuhan citizens employed metaphors to communicate about their lived experiences of the Corona Virus Disease 2019 (COVID-19) pandemic through in-depth individual interviews. The analysis of metaphors reflected the different kinds of emotional states and psychological conditions of the research participants, focusing on their mental imagery of COVID-19, extreme emotional experiences, and symbolic behaviors under the pandemic. The results show that multiple metaphors were used to construe emotionally-complex, isolating experiences of the COVID-19 pandemic. Most metaphorical narratives were grounded in embodied sensorimotor experiences such as body parts, battling, hitting, weight, temperature, spatialization, motion, violence, light, and journeys. Embodied metaphors were manifested in both verbal expressions and nonlinguistic behaviors (e.g., patients' repetitive behaviors). These results suggest that the bodily experiences of the pandemic, the environment, and the psychological factors combine to shape people's metaphorical thinking processes.

## Introduction

Metaphor, as a fundamental way of thinking, has been systematically used to understand a wide range of human experiences such as time, causation, events, emotions, the self, morality, and disease [1–6]. It has been widely acknowledged in cognitive psychology and psycholinguistics that metaphors (e.g., "death is sleep") have a conceptual dimension. Conceptual metaphors have been extensively employed in using and understanding figurative language [2, 3, 5, 6]. As metaphors can provide a framework for thinking about abstract concepts as well as complex social and health issues by means of drawing on structured knowledge from a semantically unrelated domain, metaphors seem particularly important in communication and cognition because they express, reflect, and reinforce different ways of making sense of various aspects of human lives [7] (p. 625) [8].

Previous works have shown that when people experience difficult or painful emotions or situations (e.g., cancer, pregnancy loss), they employ metaphors to understand and express

**Funding:** This study was supported by the first author's grants: Humanities and Social Sciences Research Project of Chongqing Education Commission (21SKGH143) and Foundation of First-class Discipline of Foreign Languages & Literature, Chongqing (SISUWYJY202104). There was no additional external funding received for this study.

**Competing interests:** The authors have declared that no competing interests exist.

**Abbreviations:** COVID-19, refers to Corona Virus Disease; SARS, represents Severe Acute Respiratory Syndrome; AIDS and HIV, are the abbreviated forms of Acquired Immunodeficiency Syndrome and Human Immunodeficiency Virus, respectively; MIP and MIPVU, denote metaphor identification procedure and metaphor identification procedure of VU University Amsterdam, respectively; 4E, approach to embodied cognition means embodied, embedded, enacted, extended approaches.

personal experiences [3, 7, 9–14]. For example, Gibbs & Franks [3] analyzed metaphors in six women's narratives of cancer experiences. The results showed that metaphors predominately originated from ordinary embodied experiences relating to the healthy body, such that patients concentrated on certain metaphorical ways of viewing their illnesses unique to their individual experiences [15]. Semino et al. [13] compared written data online from 56 cancer patients with their health professional controls. Patients used both Violence metaphors and Journey metaphors significantly more frequently than did health professionals. Notably, patients' Violence metaphors and Journey metaphors express both positive and negative emotional feelings of the framing contexts. Semino et al. [7] reemployed the "cancer "data in Semino et al. [13] to explore the violence-related metaphors at the conceptual level, scenario level, and linguistic level. Such multiple-level analyses of cancer metaphors have markedly extended the framing function of metaphor in cognition, discourse and healthcare practice. However, it seems that Semino et al.'s [7] study did not resolve how to identify the metaphorical symbolic activities or non-linguistic behaviors among the three levels of metaphors. To remedy this issue, Littlemore & Turner [10, 11] conducted semi-structured interviews with women who experienced pregnancy loss. It was found that metaphor as a meaning-making phenomenon extended to the image (e.g., popcorn), symbolic behavior (e.g., beer-drinking), and ascribed meaning (e.g., a baby you know to be dead still needs "parenting").

Taken together, previous studies have shown that metaphor provides important intellectual and linguistic tools for communication about intense emotional experiences. In this respect, metaphor can not only reflect past experiences, but also regulate the ways that people view their present experiences and project their future ones [3] (pp. 140–141) [16].

Along with research inquiries into metaphor in health discourses surrounding severe emotional experiences, metaphor in pandemic disease contexts has received considerable attention. For example, Wallis & Nerlich [17] examined how metaphors were used in the UK media's coverage of Severe Acute Respiratory Syndrome (SARS). Analyzing the data from five major UK newspapers, they investigated how the reporting of SARS was metaphorically framed, and how to relate the metaphorical frame of SARS to media, public and governmental responses to the disease. The results showed that the metaphor "SARS as Killer" was systematically used to illustrate the mental image of SARS, its impact on locals, and individuals' responses. The institutional and national impact of SARS and the evoked responses were framed through a bureaucratic discourse of management and a "balance metaphor" of controlled versus uncontrolled. Struggle metaphors were also extensively employed to depict the human and economic impacts of SARS (p. 2637). Since the outbreak of COVID-19, a growing body of literature has started examining how metaphors were exploited to frame the image of the COVID-19 pandemic [18–25]. Among the metaphorical conceptualizations of COVID-19, the most frequently used metaphorical image is 'War'. For instance, Rajandran [22] explored War metaphors, Journey metaphors, Direction metaphors and Liquid metaphors in public speeches of the presidents of Malaysia and Singapore. The results suggested that War metaphors showed the highest ratio, given that the frame of war could reveal people's responses to the COVID-19 pandemic, reflecting the impact of COVID-19 on multiple aspects of people's lives. In that sense, the frame of war metaphors could raise people's awareness of COVID-19 [22]. Nevertheless, War metaphors have been criticized for inappropriately depicting COVID-19 as a malevolent opponent, creating excessive anxiety, potentially legitimizing authoritarian governmental measures, and implying that dead people did not fight hard enough [24] (p. 50). In other words, War metaphors have at times been regarded as raising anxiety, destroying the unity and creating hostility among people, and even deteriorating relations between politicians and the public [21, 23]. Moreover, the use of War metaphors in framing social issues is at times blamed for simplifying the complicated issues into dichotomous conflicts, given that it

draws public attention to external threats rather than to failed policies, healthcare problems, inequities and injustice. Thus, War metaphors may negatively influence poor or vulnerable groups to some extent [18]. In healthcare practice, the use of War metaphors may increase healthcare workers' fears because patient care may conflict with the self-care for healthcare workers [20, 21].

Regarding other metaphorical conceptualizations of COVID-19, Craig [19] compared the metaphorical frames of Acquired Immunodeficiency Syndrome (AIDS) and the COVID-19 pandemic with Military metaphors, Freedom metaphors, and Light metaphors. The results showed that metaphors about COVID-19 highlighted that the pandemic was shared by all people, whereas metaphors about AIDS were employed to stigmatize patients infected with Human Immunodeficiency Virus (HIV). Wicke & Bolognesi [25] investigated how people employed the War metaphors, Monster metaphors, Storm metaphors, Tsunami metaphors, and the literal frame of "family" in online writing when talking about COVID-19. The research findings revealed that although the predominate metaphor usage was War, the literal frame of "family" was the most frequently used pattern. Furthermore, among multiple topics in online writing, War metaphors were only used with certain topics. Other metaphorical frames need to be identified to present different dimensions of COVID-19 [25]. Most recently, Semino [24] drew on a large corpus of news and found out that the use of Fire metaphors was more frequent than that of War metaphors concerning various facets of the COVID-19 pandemic.

Clearly, SARS and COVID-19 offered an opportunity to explore the metaphorical framing of the pandemic crisis. Previous research has shown that a wide variety of metaphors have been used in communication about the COVID-19 pandemic in the context of news discourse, political discourse and online writing. To date, however, very little empirical research has been conducted to address how metaphors are facilitated in people's narratives of their lived experiences during the COVID-19 crisis (e.g., [26]). Notably, Stanley et al. [26] conducted in-depth semi-structured interviews with 44 American individuals. Participants were asked to compare the COVID-19 pandemic with an animal and a color, and to explain their metaphors. The metaphor analysis uncovered four convergent mental models regarding participants' experiences of the pandemic, namely, Uncertainty, Danger, Grotesque, and Misery. Furthermore, grief, disgust, anger, and fear were associated with the four mental models. The results suggest that metaphor can indeed conceptualize individuals' deeply felt, implicit emotions about their lived experiences of the COVID-19 pandemic. In the present study, we sought to investigate how a group of Wuhan citizens (N = 27) employed metaphors to talk about their emotionally-complex, isolating experiences of the COVID-19 pandemic through in-depth semi-structured interviews. Free conversation and an elicited metaphor approach were employed to resolve the limitation of "forced metaphor" [26]. The data analysis captured participants' different kinds of emotional states and psychological conditions during the pandemic. An assumption of embodied cognition is that "the mind is not simply in the head, but is organismically embodied, contextually embedded, and environmentally extended and distributed" [11] (p. 47) [27] (p. 161). Hence, the metaphors examined in this study are not merely limited to linguistic metaphors; the use of images, symbols, behaviors, and actions are also viewed as important means to construct metaphorical cognition [10, 11] (p. 47). Essentially, we argue that metaphor can provide functional and concrete cognitive patterns in articulating and constructing participants' subjective lived experiences, offering in-depth insight into their emotional states, identifying personal mental health issues and ultimately promoting positive changes. The hope is that the present paper will serve to shed light on the ways that metaphor can be employed by mental health professionals when evaluating and measuring people's mental health before implementing effective mental health support and promoting accurate health awareness in the fight against the pandemic crisis.

## Materials and methods

### Participants

A total of 27 adults (twelve men, fifteen women), between the ages of 19 and 71 (mean age = 30.3 years), participated in a telephone interview as paid volunteers. They were contacted through an online recruitment advertisement by convenience sampling. The recruitment advertisement was posted in the Wechat social networking app. All participants lived in Wuhan during the lockdown, except for two COVID-19 patients who lived in Wuhan but had received treatment in Xiangyang, Hubei Province. Written informed consent was obtained from all individual participants before the interview. Among the 27 participants, three were diagnosed with the COVID-19 infection and admitted to the COVID ward, four were healthcare workers supporting COVID-19 patients in frontline hospitals, four were social workers who had offered help in Wuhan, nine were healthy college students staying at home during the lockdown, three were company staff, one was a common pneumonia patient, one was the family member of a COVID-19 patient, one was a teacher, and one was a journalist who had conducted interviews and written news reports in Wuhan during the pandemic. See Table 1 for participants' sociodemographic characteristics. The study was approved by the Ethics Committee of the authors' institute. All procedures involving human participants were in accordance with the ethical standards of the institutional and national research committee and with the Helsinki declaration and its later amendments or comparable ethical standards.

**Table 1. Socio-demographic characteristics of participants (N = 27).**

| Characteristics | Number (%) |
|---|---|
| **Gender** | |
| Male | 12 (44.4%) |
| **Age** | |
| 19–25 | 14 (51.9%) |
| 26–30 | 2 (7.4%) |
| 31–40 | 7 (25.9%) |
| >40 | 4 (14.8%) |
| **Occupation** | |
| College Student | 9 (33.4%) |
| Frontline Healthcare Professional | 4 (14.8%) |
| Social Worker | 4 (14.8%) |
| COVID-19 Patient | 3 (11.1%) |
| Company Employee | 3 (11.1%) |
| Family Member of a COVID-19 Patient | 1 (3.7%) |
| Common Pneumonia Patient | 1 (3.7%) |
| Journalist | 1 (3.7%) |
| Teacher | 1 (3.7%) |
| **Education** | |
| With a University Degree | 24 (88.9%) |
| Without a University Degree | 3 (11.1%) |
| **Marital Status** | |
| Married | 10 (37%) |
| Unmarried | 16 (59.3%) |
| Bereft of One's Spouse | 1 (3.7%) |

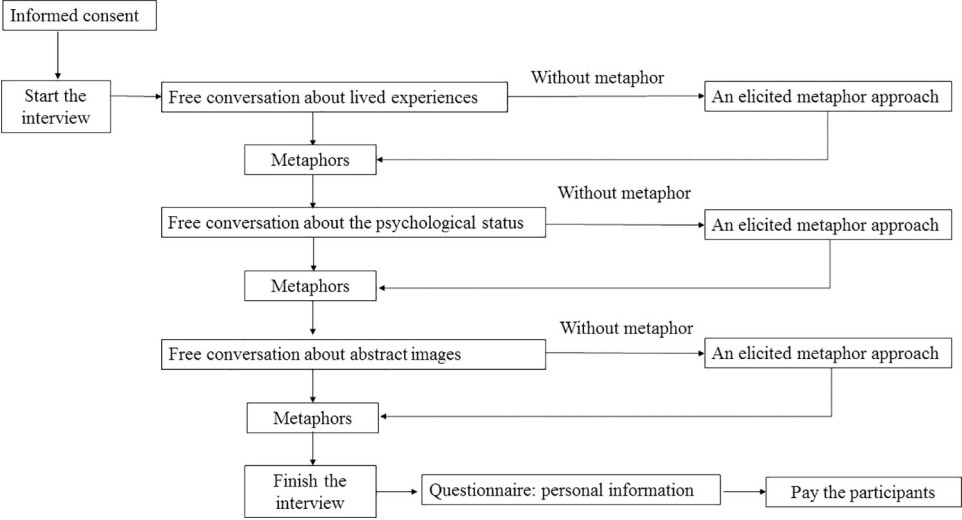

**Fig 1. Data collection procedure.**

## Interview procedures

The semi-structured interviews were conducted by cellphone or social networking WeChat phone. Each interview lasted about 40 minutes and was audio-recorded. We employed free conversation [3, 9, 10] and an elicited metaphor approach [28] to collect pandemic narratives. See Fig 1 for the interview processes.

For the free conversation narratives, we asked participants to recall their lived experiences and feelings concerning the COVID-19 pandemic in Wuhan. Interview questions included multiple dimensions of their life experiences during the pandemic, such as "What did you do at the outbreak of COVID-19?", "How did your work and daily life were affected by the pandemic?", "What gave your comfort when you felt frustrated?", "What changes in perceptions of the world did you have?", "How did you feel in witnessing death every day?", "How did you feel when the city was locked down?", and "What was the most unforgettable event you experienced during the pandemic?".

Since the answers to the free conversation questions might not yield metaphors, two over-arching questions were added to elicited metaphors: "Please describe X in a nonliteral way" or "What was X like". The "X" could be a scenario, an event, a specific feeling, a place, or an object. The patterns of the prompts were flexible, such as "What was the coronavirus like in your mind?", "What was self-isolation like?", "What were the hospital and city like in the pandemic?", "What was death like during the pandemic?", "What were your feelings like concerning the lockdown of Wuhan?".

The free conversation and elicited metaphor method complemented each other in two ways. First, the free conversation method could ensure that participants were not conscious of the metaphorical nature of their narratives, thus could overcome the problem of "forced metaphor" [26]. Second, the following elicited metaphor method could guide participants to produce adequate metaphors pertaining to their lived experiences of the pandemic. The interviews were conducted from June 2020 to July 2020 by one of three authors specializing in psycholinguistics.

## Data coding

The audio recording lasted 18 hours and 11 minutes. The interviews were transcribed verbatim by the individual interviewer. A sample of 262,296 Chinese characters from the corpus was

manually coded for metaphorical expressions following the Metaphor Identification Procedure (MIP) by Pragglejaz Group [29]. Specifically, an utterance was coded as a metaphor when its "contextual meaning" contrasted with a more physical and concrete literal meaning, and where the contextual meaning was understood via a comparison with the literal meaning, as illustrated in the use of "neighbor" in the expression "During COVID-19, life and death were neighbors". Furthermore, adopting Steen et al.'s definition of "direct metaphor" within the Metaphor Identification Procedure of VU University Amsterdam (MIPVU) [30], in the present study metaphorical expressions included other figurative comparisons such as simile. Each metaphorical utterance was further allocated to a semantic field such as Battling, Violence, Journey, Game, Weight, Temperature, Light and Darkness on the basis of its basic meaning [7, 10, 11]. For utterances where one metaphor was assigned to more than one category, we adopted a maximally inclusive approach in which all categories were treated at the same level [11] (p. 50). For instance:

> With further understanding of the patients, I considered myself as a brave warrior, who must overcome all the difficulties to cure them. When I charged at COVID-19 like a female warrior, my blood was heated. (Subject 23, doctor)

The doctor's statement above was annotated under the categories of War metaphor and Liquid-based metaphor. Next, we identified the topics of each metaphorical utterance and source-target domain mapping. For instance, the topic of the above metaphorical utterance relates to "combat COVID-19". The target domain of the War metaphor is curing the COVID-19 patients in the hospital, and the source domain is fighting on the battlefield. The source domain of the liquid-based metaphor is a heated fluid in the body, and the target domain is enthusiasm for working in the COVID-19 hospital.

The metaphorical coding was conducted by two independent coders. The first coder coded all the metaphor categories, identified the topics, and clarified the source-target domain mapping of each metaphor. To establish the reliability of coding the metaphorical expressions, the second coder independently coded 100% of the entire data set. The degree of reliability was high (Kappa = 0.97); disagreements were resolved via discussion until both coders reached a consensus. All metaphors were translated from Chinese into English.

## Results

370 metaphorical utterances were coded from the 27 participants' narratives. The 370 metaphors did not all represent distinct metaphorical concepts as many were similar across the 27 narratives. 49 metaphor categories were identified around 70 topics.

We used the word cloud software Wordle to visualize the distribution of token frequency of the 49 metaphor categories and related topics. The clouds offer prominence to keywords that appeared more frequently. Fig 2 presents the visualization of the 49 types of metaphor (the examples are listed in S1 Appendix). Fig 3 illustrates the topic matrix of the 370 instances of metaphorical utterances (also see S2 Appendix).

It can be seen from Fig 3 that the salient topics mentioned included the mental image of COVID-19, the emotional state at the outbreak of the pandemic and isolation period (e.g., anxiety, fear, panic, grief, nervousness), life and death, the lockdown of Wuhan city and home quarantine, combatting COVID-19, contribution by nation and medical staff, people's extreme feelings, behaviors, and dreams, as well as social relationships between the individual, health professionals, the family, and the nation.

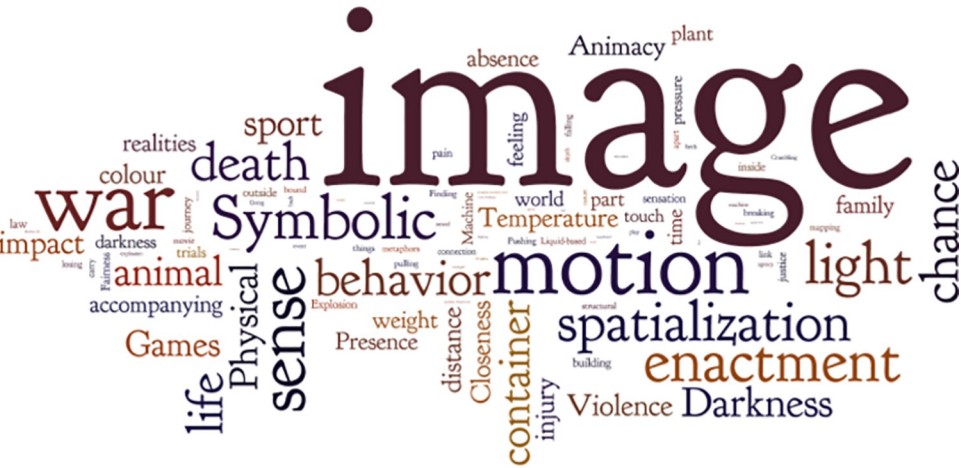

**Fig 2. Overall distribution of the pandemic metaphors.**

Looking at Fig 2, the 27 participants used a wide range of metaphors to narrate different aspects of their COVID-19 experiences related to image, motion, war, sense, symbol, light, spatialization, life, death, containers, animals, games, impact, violence, animacy, temperature, and color. The size of words in Fig 2 corresponds to the relative frequency of the metaphor categories, in that one metaphorical utterance may encompass more than one metaphor category, thus leading to overlapping categories among the 370 metaphorical utterances (also see S1 Appendix). The ten most frequent patterns were Image metaphors (e.g., "Doctors are angels/city heroes"), Motion metaphors (e.g., "My heart suddenly jumped out"), War metaphors (e.g., "The hospital was a battlefield"), Sensory metaphors (e.g., "I could not smell the happy atmosphere of the spring festival"), Symbolic metaphorical enactment (e.g., "A frontline doctor substituted the 'profile photo' of his social networking apps with 'the picture of a well-known coronavirus expert' before entering the isolation ward"), Light metaphors (e.g., "I felt the sunshine of life when I left the hospital, where the darkness of disease was spread"), Spatialization metaphors (e.g., "I fell into a panic"), Life metaphors (e.g., "Life is a bubble"), Death metaphors (e.g., "Death means the soul rises to heaven"), non-verbal metaphorical behaviors (e.g., "A

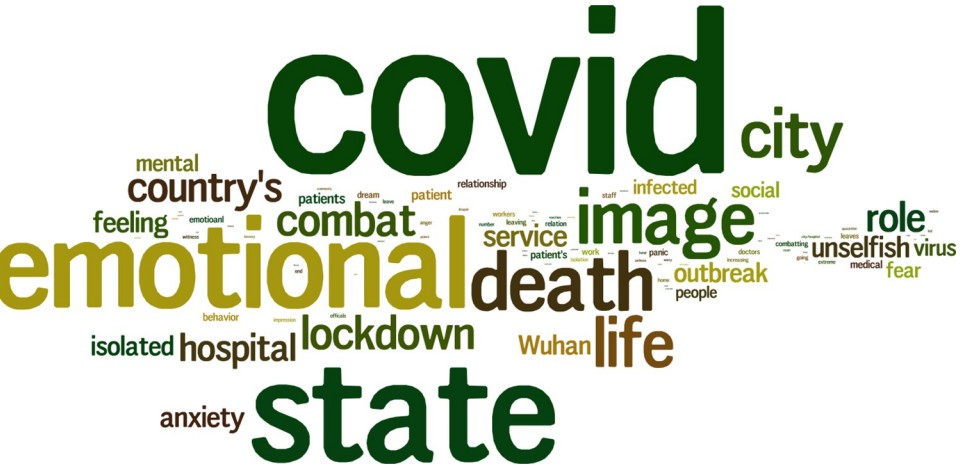

**Fig 3. Topic matrix of the pandemic metaphors.**

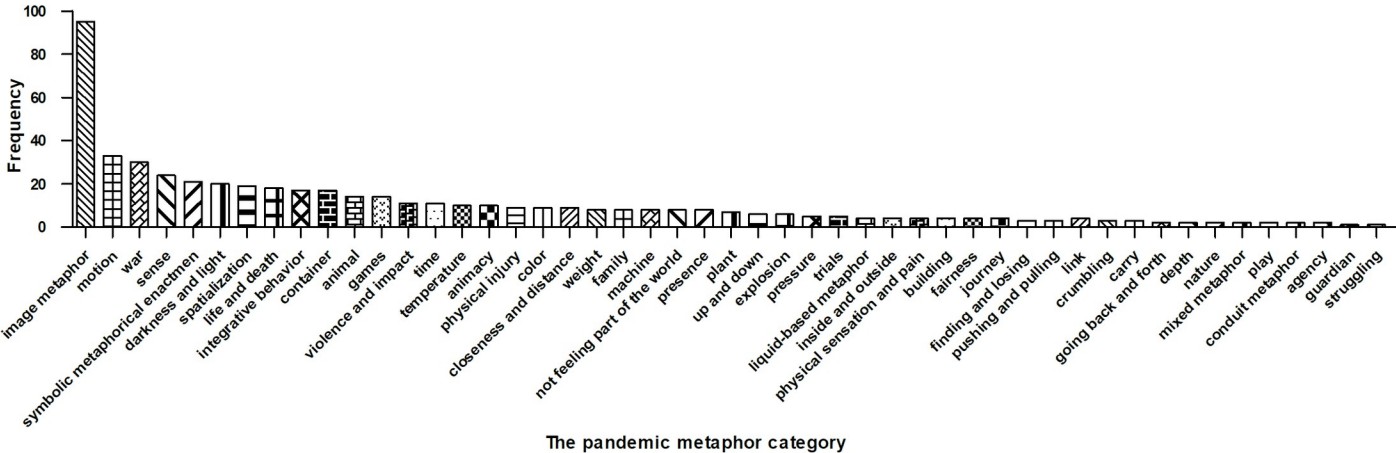

**Fig 4. The frequency distribution of the 49 metaphor categories.**

coronavirus patient wrote a will, tidied up the house, and lay in bed waiting for death, showing a physical ritual of coming to terms with the potential end of life"), and Container metaphors (e.g., "The city became empty"). See Fig 4 for the frequency distribution of the 49 pandemic metaphor categories.

It is also worth mentioning that multiple metaphors were used to depict a single concept. For instance, a pneumonia patient described his view on death as follows:

*Death is returning. It is like a deep sleep. It is like dust fading away in the air. It is like a drop of water flowing into the sea, disappearing forever. (Subject 1, a common pneumonia patient)*

Here, ideas about death during the COVID-19 pandemic were conceptualized as four different metaphors: DEATH IS RETURNING, DEATH IS SLEEP, DEATH IS FLYING DUST, and DEATH IS A DROP OF WATER. The informant offered different entailments to express self-perception of his disease. Although the multiple metaphors appear to be irrelevant to each other, the mixed metaphors can reveal the multiple ways he perceived certain experiences at different times [3] (p. 157). Hence, it seems reasonable to conclude that mixed metaphors are valuable in situations concerning personal emotional intensity and other extreme experiences [31], particularly for patients who have been exposed to a traumatic stress or catastrophic events during the pandemic.

At a general level, participants can express their lived experiences of COVID-19 via conventional and conceptual metaphors. Thus, the metaphors that were grounded in their embodied experiences were employed to understand lives disrupted by the COVID-19 crisis (e.g., DIFFICULTY IS WEIGHT; AFFECTION IS WARMTH; HAPPY IS UP, SAD IS DOWN; HEALTH AND LIFE ARE UP, SICKNESS AND DEATH ARE DOWN; EMOTIONS ARE NATURAL FORCES; LIFE IS A JOURNEY/STRUGGLE).

The findings also show that participants occasionally generated novel metaphors when talking through their extreme emotional experiences. New inference patterns were used to extend the mapping of a specific conceptual metaphor [6] (p. 67). For example, a female coronavirus patient described her emotional state when she had to leave her lovely daughter and went to hospital:

*The feeling of being reluctant to leave my daughter is... when you play a computer game, you have to keep upgrading, through which you get a lot of equipment. Just like the equipment, I*

*bring up my daughter and nurture her step by step. But suddenly one day, when you have to abandon all the equipment, you feel very regretful and extremely distressed. (Subject 16, coronavirus patient)*

The woman's narrative captured her sadness when leaving her little daughter. It reveals the woman's experience from the onset of the COVID-19 illness to self-isolation and hospitalization. Playing a computer game vividly visualized her emotional state at leaving her beloved daughter. Here, her daughter was metaphorically construed as victory awards in the computer game. Losing the victory awards in a computer game is thus mapped onto leaving her daughter, as an upsetting event for a mother. This novel metaphor can be seen as the extension of the conceptual metaphor LIFE IS A GAME, effectively reflecting the coronavirus patient's extreme emotional state under the COVID-19 pandemic.

## Discussion

The present study aims to reveal the ways that metaphor is used to express the emotionally-complex, isolating experiences of the COVID-19 pandemic. The data suggested that a large variety of metaphors were spotted in participants' narratives, and the extreme emotional and psychological experiences yielded a rich set of metaphorical mental images of the crisis. Furthermore, most metaphors on individuals' emotional states were closely associated with personal embodied sensorimotor experiences. In addition, metaphors were manifested not only in verbal expressions but also in nonlinguistic behaviors and other symbolic means [6] (p. 57). This section concentrates on the metaphorical thinking processes regarding participants' mental, emotional, and psychological experiences of the COVID-19 pandemic.

### Mental imagery of COVID-19

Metaphors can increase the comprehension of the conventional mental imagery via cross-mapping between sensorimotor and subjective experience domains [6] (p. 45) [24]. As one interview question concerned participants' perceptions of COVID-19, metaphors were extensively employed when participants depicted their mental imagery of the disease. Animal, Light and Darkness, Season, Animacy, Ghost, Devil, and Doomsday were the most commonly adopted mental images associated with individuals' subjective experiences of the COVID-19 pandemic. With respect to the Animal metaphors, COVID-19 was conceptualized as a beast, a bat, a thorny hedgehog, or a deer in the woods. In reality, a beast can attack human beings; a bat operates in darkness; and a deer in the deep forest is mysterious. The Animal metaphors map properties of animals onto participants' mental representations of COVID-19, highlighting the risk and mystery of the disease. Via the Light and Darkness metaphor, the outbreak of COVID-19 was metaphorically conceptualized as the mental image of darkness. The metaphorical conceptualization revealed the horrible nature of COVID-19. Regarding the Seasons metaphor, COVID-19 was construed as the image of a no-man's land in deep winter filled with vacant property and stark trees. Such an image transforms participants' perception of Wuhan city into an isolated COVID-19 epicenter during the times of lockdown. The mental image of a ghost, devil, and evil person in understanding COVID-19 is deeply rooted in participants' real-world experiences of isolation, lockdown, and death in Wuhan city. Notably, participants were very likely to draw on the doomsday images in movies, indicating their mental representations of COVID-19. In doomsday scenarios, the earth is destroyed; humans are killed; cities are isolated; and the air is poisonous. The inference pattern of a doomsday scenario was mapped onto the image of COVID-19, suggesting that the pandemic has, unsurprisingly perhaps, posed multiple psychologically stressful challenges and may even increase the

risk of mental health problems. The rich image metaphors for COVID-19 are not arbitrary. The meaning of these images is motivated by the metaphorical mapping and certain conventional mental images. In our data, each metaphorical utterance of COVID-19 comes with a conventional mental image and knowledge about that image (e.g., animal, light and darkness, season, ghost, devil, and doomsday). A conventional metaphorical pattern maps that source domain knowledge onto target domain knowledge of COVID-19 (see [6] (p. 68), [24]). Different metaphorical patterns profile informants' multiple emotional responses to the crisis (e.g., dangerous, mysterious, horrible, isolated feeling). Given that most mental images of COVID-19 can be shared across a large portion of the speakers within the language and culture (i.e. Chinese), the conventional images of COVID-19 in our data somehow reflect part of people's long-term memory [1, 6] (p. 69).

## Embodied nature of metaphor in emotional experiences

As part of the data collection, we asked the overarching question "How did you feel" after participants reported a detailed event. The results showed that metaphors were extensively employed to show participants' intense emotional reactions to the COVID-19 pandemic. Specifically, more elicited metaphors in the study were exploited to report how informants responded to specific pandemic-related events than those merely used to describe the events in the same scenario. This is consistent with the previous literature regarding the association between metaphor and emotional experiences. That is, people at times draw on multiple metaphors to describe emotion-focused experiences [2, 9, 11, 12, 14, 32].

An important observation about emotional metaphors is that many of their source domains reflect patterns of bodily experience [6]. Participants construed emotion through sensorimotor experiences by their body parts and different sensory media (e.g., vision, sound, touch, taste, and general kinesthetic actions of the entire body). The varied bodily experiences gave rise to embodied concepts for intense emotion under the COVID-19 pandemic. In the following two examples:

> *For me, it is an unexpected disaster. It was like I was smashed by a stone falling from the sky, I felt extremely irritated at the time. (Subject 16, coronavirus patient)*

> *I hadn't been separated from my daughter for such a long time. . .it was like cutting the flesh from my body. (Subject 16, coronavirus patient)*

The patient metaphorically described her negative experiences of being infected by coronavirus in terms of physical injuries. Infection was conceptualized as being smashed by a huge stone; leaving her daughter was depicted as cutting flesh from her body. Drawing on the primary metaphors EMOTIONAL DIFFICULTY IS PHYISICAL INJURY and PSYCHOLOGICAL EFFECT IS PHYSICAL CONTACT, the two metaphorical utterances map injuries and pain in the body onto the pain in the psychological experiences and states of mind. These physical experience-related metaphors are highly embodied and prevalent when people are reporting emotionally intense experiences during the pandemic. From the patients' point of view, the embodied metaphors at the physical level can help them to come to terms with their disease [3].

A similar scenario at the physical level was described by a doctor in a COVID-19 emergency room. She reported her feelings when she came out of the COVID ward:

> *Leaving the COVID-19 in-patient ward. . .I felt like my body had been "hollowed out" and I was eager to catch my breath. I remember that when the COVID-19 pandemic was over, I*

*slept for a whole day on the first day and I was reluctant to move and felt really relaxed. (Subject 23, doctor)*

Again, the doctor placed herself at the center of an embodied metaphor which draws on primary metaphors such as THE BODY IS A CONTAINER, PSYCHOLOGICAL HARM IS PHYSICAL INJURY. This embodied metaphorical scenario reveals the doctor's nervous, anxious, stressful state of mind under the pandemic.

Apart from the emotion metaphors tied to human body parts, enduring embodied experiences such as battling, hitting, weight, temperature, motion, violence, and light were extensively projected onto participants' intense emotional states under the COVID-19 pandemic. For instance:

*I guess his parents' death left him with a deep psychological trauma. I believe that, that night, his mental line of defense was totally smashed without any preparation. (Subject 14, doctor)*

In this metaphorical scenario of battling, the doctor described the emotional state of despair and grief of a coronavirus patient who witnessed the death of his parents in an emergency COVID-19 hospital. Indeed, his whole family was infected by COVID-19. Here, the scenario appears to be underpinned by several different embodied metaphors, including: EMOTIONAL DIFFICULTY IS PHYISICAL INJURY, DIFFICLTIES ARE OPPONENTS, KEEPING PEACE WITH COVID-19 IS VIOLENT FIGHTING, EMOTION IS AN OPPONENT, and EMOTION IS A PHYSICAL FORCE. These combined metaphors uncover the patient's extreme mental state of despair and grief when all his family members as well as he himself were infected by the virus.

Strong emotional experiences can also be framed by weight metaphors, as the following examples illustrate:

*I am now suddenly feeling heavily-laden. What comes to my mind is all about the ambulances downstairs. They took away one patient after another. We didn't know whether these patients could come back or not. When I think about it, I feel heavily-laden. (Subject 15, teacher)*

*With the increasing mortality rate during the serious pandemic, I felt weighted down, as if a heavy stone was crushing me. I felt extremely worried and weighted down. (Subject 17, college student)*

The first scenario concerns the fear and pressure when witnessing the fact that a growing number of coronavirus patients were sent to the hospital. The second scenario refers to the anxiety at hearing the news of the COVID-19 mortality rate. The extreme fear, stress, and anxiety are underpinned by the embodied primary metaphors DIFFICULTIES ARE BURDENS. Here, the subjective experience is difficulties in emotion, and the sensorimotor domain is muscular exertion [6] (p. 50). The discomfort or disabling effect of lifting or carrying heavy objects (i.e. a stone) is projected onto the pressure, fear, and anxiety under the serious situation of the pandemic.

The Temperature metaphor reflects people's physical and sensory experiences. For instance:

*They felt chilled; the city was over, so was I. It's a simple truth. (Subject 27, social worker)*

*When all kinds of sad and negative news were surging into you, you might feel so cold and trembling with fear, as if doomsday was coming. (Subject 6, company staff)*

The two examples above are about the feelings concerning the outburst of the COVID-19 pandemic. Participants used the embodied metaphors AFFECTION IS WARMTH and DIFFI-CULTIES ARE COLDNESS to show their negative emotional experiences. Here, the sensori-motor domain is temperature, and the primary experience is that people feel warm while being held affectionately. These embodied experiences are projected onto the bad emotion at the outbreak of the COVID-19 pandemic, formulating the metaphor NEGATIVE EMOTION IS COLDNESS.

In previous literature, journey metaphors are frequently used to conceptualize embodied experiences of illness [3, 13]. Below is an instance of conceptual metaphor in the narratives of a woman who was infected by the coronavirus:

*The moment I got into the ambulance, it was my departure for death. (Subject 19, coronavirus patients)*

The conceptual metaphors LIFE IS A JOURNEY and DEATH IS DEPARTURE exhibit the woman's feeling of despair and despondency. In the metaphorical projection, illness in one's life corresponds to the obstacles along life's journey. The infection of COVID-19 is conceptualized as stepping on a path to death. In this scenario, as the woman understood her situation of illness in an emotionally negative manner, her ordinary embodied experiences in getting into the ambulance is mapped onto traveling to death.

Emotions and spatialization are closely intertwined. Consistent with the previous research on spatial metaphor in communication about depressed and anxious events [10, 11], the present data yielded embodied metaphors that are very spatial, as shown in the following examples:

*From the beginning, when the number of infected people increased greatly every day, to later when the number was under control, our emotional state had been fluctuating all the time; it began with a sigh, then turned to worry and fear, and finally to hope and victory. It was like a roller coaster, which could push you down all of a sudden, and the journey of your emotions was up and down like a roller coaster. (Subject 14, doctor)*

*Some doctors came, and my anger rose up so that I yelled at the people who came to pick us up. (Subject 19, coronavirus patient)*

The first scenario (Subject 14) was described by a frontline doctor in Wuhan. He experienced various stages of the COVID-19 pandemic. The statement involves the common embodied spatial metaphors HAPPY IS UP, SAD IS DOWN, and CHANGE IS MOTION. His metaphorical emotional experience of COVID-19 as the "up and down of a roller coaster" reveals his complex emotional state, from worry and fear to victory and happiness. Illustrating a Covid-19 patient's anger at going to hospital in the second scenario (Subject 19), the embodied metaphors were used in a slightly different way. His narratives of angry emotion as rising up in the body are based on the embodied metaphor EMOTIONAL CHANGE IS MOTION and ANGER IS THE HEAT OF FLUID IN A CONTAINER. His anger metaphor involves two primary metaphors ANGER IS HEAT and THE BODY IS A CONTAINER for the emotions. Here, the source domains (HEAT and CONTAINER) are compatible, given that substance can be inside containers. The target domains (ANGER and BODY) can also be combined, because human bodies can show signs of anger [31]. In Chinese culture, the spatial orientation of the heated anger is construed as moving up, as opposed to the primary metaphor HAPPY IS UP. The instance suggests that compound embodied metaphors are seldom the same across

languages and cultures [31]. The cultural variation in structuring embodied metaphor is related to specific embodied experiences of the pandemic, as one more example shows:

> *Going in and out of the isolation ward was like taking an elevator. You might feel oppressed when you went up, especially when you got higher and higher, and then you reached the peak. Assuming that I was in a transparent elevator, I wondered what would happen if I fell down when getting higher and higher. As I reached the peak, the fear decreased with more companions around. Later, the elevator descended slowly, during which you might surprisingly lose gravity. The elevator was going down and the pandemic was getting better. For example, there were no more confirmed cases, or the number of confirmed cases decreased slowly until no new cases appeared. Finally, the door of the elevator opened and I left the COVID-19 ward. Arriving at the destination, I stepped safely onto the ground. (Subject 20, nurse)*

This scenario was narrated by a nurse in a COVID-19 hospital in the isolation period. Her emotional state of entering and exiting the coronavirus patients' ward was construed as taking the elevator. In the ascending period, experiencers may feel oppressed physically. The higher the elevator rises, the more oppressed they may feel. In contrast, the fall of the elevator would make experiencers feel relaxed, although they may initially go through a period of losing gravity. This scenario conflates embodied metaphors, including LINEAR SCALES ARE PATHS, CHANGE IS MOTION, MORE IS UP, LESS IS DOWN. Here, the primary emotional experience regarding the changing quantity of infected patients is grounded in our bodily experience of vertical orientation. The rise in the number of infected patients corresponds to upward motion, while the fall in the number of infected patients is downward motion. It is noteworthy that the novel emotion metaphors here contradict the primary spatial metaphor HAPPY IS UP, SAD IS DOWN. Rather, moving up is projected onto the negative emotion of oppression, whereas moving down is mapped onto the positive emotion of feeling safe.

In summary, emotion metaphors in the present study were embodied through bodily experience in the world. These metaphors associated sensorimotor experience with the subjective experience of the COVID-19 pandemic. The source-domain logic (e.g., body parts, battling, weight, temperature, spatialization, violence, and light) arises from the inferential structure of the sensorimotor system [6]. Clearly, the source domain inference patterns of embodied metaphors reveal both universal and cultural variation in line with the specific subjective experiences, building a foundation for understanding the emotional experiences of the COVID-19 pandemic.

## Metaphorical enactment in symbolic behaviors

According to the embodied, embedded, enacted, and extended (4E) approach of embodied cognition, metaphor can be something we 'do' and not limited to something that simply exists [9] (p. 15). In the interview, metaphors were not only detected in linguistic expressions, but also enacted in symbols and integrative behaviors [6] (p. 57). The metaphorical symbolic behaviors can reveal participants' extended emotions (e.g., anxiety, fear, despair) of their lived experiences during the COVID-19 pandemic. For example:

> *She said she wrote a will at home. After tidying up the house and cleaning herself up, she lay down on the bed with the will, waiting for death at home. (Subject 14, doctor)*

> *I worried about my 5-year-old daughter. I even finished a will, stating how to distribute my personal property if I died from COVID-19. (Subject 16, coronavirus patient)*

Here, the integrative behaviors were reported using the third-person and first-person perspectives respectively about two patients. Both extracts described the action of writing a will when death was approaching in the near future. By drafting a will when infected by COVID-19, the two patients reconciled their emotional states of panic, fear, grief, and anxiety between the distant future and the present—being on the fringe of extreme emotional despair. The behaviors of writing a will and cleaning up can be regarded as an instance of metaphorical enactment of time compression [11] (pp. 57–58). Specifically, the patient's behavior is a physical ritual of coming to terms with the potential end of life in which it brings a salient moment from a far-off future of death into the present. The metaphorical comparison occurs between a reality of being infected by COVID-19 and the end of life in the future.

The metaphorical symbolic behaviors also occur in the case of doctors under the extreme pressure of working in the emergency COVID-19 hospital. One interesting example is that one doctor replaced the head image of his WeChat (the most popular social networking program in China) with a picture of coronavirus expert Nanshan Zhong:

> *I was petrified of death, really petrified of death. Before I entered the COVID-19 ward, I replaced the profile photo of WeChat with a photo of Dr. Zhong Nanshan, hoping for his blessing. (Subject 14, doctor)*

Doctors were exposed to coronavirus patients in an isolated space and witnessed the death of patients every day. The extreme emotional experiences of fear, anxiety, panic, and pressure during the COVID-19 pandemic enabled them to associate the symbol of the head image of the COVID-19 expert Nanshan Zhong on social networking as a way to seek inner peace and blessings. This is a metaphorical integrative behavior extending the doctor's emotional state in the hospital.

Another remarkable symbolic metaphorical enactment is the repetitive behaviors of coronavirus patients who showed the traits of obsessive-compulsive neurosis due to their extreme anxiety. The following two examples of coronavirus patients' behaviors were reported in terms of a third-person perspective by the frontline healthcare professionals:

> *The majority of patients were extraordinarily anxious. . .When they were admitted to the hospital, they constantly called the doctors and nurses, and kept talking to us about the illness. Why? That's because they might feel safe with us around. (Subject 14, doctor)*

> *As long as you touched her, she became highly sensitive, even frequently making physical attacks on you. She kept holding my hand, and I tried to free my hand, but I failed. They were all extremely nervous, frequently requesting a nucleic acid test, the antibody detection, the CT scan, or something like that. (Subject 20, nurse)*

The reported behaviors of the anxious coronavirus patients in the isolation hospital demonstrate elements of the obsessive-compulsive disorder: constantly begging doctors to save their lives, frequently talking with doctors about their illness to seek a sense of security, holding nurses' hands tightly all the time, frequently requesting nucleic acid or CT tests, and even attacking the medical staff. Such repetitive behaviors reflect their anxiety, nervousness, and despair after being infected by the deadly coronavirus. According to Freud's view of obsessive-compulsive neurosis, coronavirus patients' behaviors reflect not only a repressive process that fails again and again but also a defensive move made against unconscious impulses that are threatening to invade consciousness [33]. The metaphorical aspect of patients' obsessive symptoms consists in the fact that their behaviors seem to be similar to a dream object with a

symbolic dimension: both the obsessive thoughts and compulsive rituals symbolize repressed contents that are projected outward, turning into threatening external objects while reflecting the subject's inner world and its destructive impulses. Thus, the metaphorical enactment of coronavirus patients' repetitive behaviors creates a "semantic shift" in which the threatening inner objects of thought are displaced in an external, concrete one and inverted into their opposites [34] (pp. 262–263).

## Study limitations and future research

This study has some limitations. First, we examined metaphors in the framing of the COVID-19 pandemic under a collective emotion [26]. It assumed that participants from Wuhan would have the same directions or tendencies as regards emotions or communicating emotions towards the COVID-19 experiences. Although participants were recruited from the same city, the metaphor informants were of different types. Each of their perspectives could be different with respect to their COVID-19 experiences. A second limitation has to do with the small sample size. This study involved a qualitative analysis of pandemic metaphors in a convenience sample of 27 participants from Wuhan during COVID-19. The results may not be generalizable to a wider population. Furthermore, the lack of face-to-face interviews might have prevented participants from freely expressing their possible frustrations during the pandemic [35].

In the light of the potential challenges, future research could employ large samples of video or face-to-face interviews to complement the present research. Furthermore, quantitative models can be used to distinguish different metaphorical conceptualizations of COVID-19 among the different types of participants. Individual background factors such as gender, age, and quarantine time can be powerful predictors in exploring the framing variation of metaphor in communication about the lived experiences of the pandemic.

## Conclusion and implications

In the present study, the metaphorical analysis of people's lived experience narratives about the COVID-19 pandemic has revealed that the communicative situation may enhance the likelihood of metaphors being used when depicting individuals' intense emotional and psychological experience-related scenarios. Participants employed multiple metaphors as they talked about mental images of COVID-19, extreme emotional experiences, and symbolic behaviors during the pandemic. Notably, most metaphorical narratives about experiences of the pandemic were conventional and conceptual, reflecting enduring metaphorical patterns of thought grounded in ordinary, embodied sensorimotor experiences such as body parts, battling, hitting, weight, temperature, spatial orientation, motion, violence, light and darkness [3, 6, 11, 24]. Metaphor, as a way of thinking and meaning-making, also extends from the linguistic level to the level of symbolic behavior (e.g., patients' repetitive behaviors).

The analysis of embodied metaphors in communication about the pandemic can offer clinical implications for therapists and clients for using the literal and metaphorical meanings in pandemic-related psychotherapy. As a powerful therapeutic tool, metaphor has worked reasonably well in helping people to reframe their experiences and amend their painful conceptualizations [9, 36, 37]. In the context of pandemic-related psychotherapy, therapists can uncover patients' metaphors with negative, disempowering effects and then adopt different metaphors in positive, empowering ways [13] (p. 6). For instance, therapists may use the positive images such as sunshine, a magic mirror, spring, green trees, and beautiful flowers to reconstruct clients' mental imagery of the coronavirus. Furthermore, positive embodied metaphors such as HAPPY IS UP, AFFECTION IS WARMTH, INTIMACY IS CLOSENESS,

HELP IS SUPPORT, and CONTROL IS UP can be facilitated to regulate clients' emotional experiences of the pandemic. Regarding metaphorical behaviors, therapists can guide clients to imagine happy events such as preparing a celebration party for winning the battle with COVID-19. The purpose is to project positive emotion or attitude to the pandemic in the future. Along this line of enquiry, future research can focus on COVID-19 patients, and investigate their mental health status through metaphor analysis. In doing so, metaphors can be used as a powerful tool in psychological intervention on persons infected with COVID-19.

## Supporting information

**S1 Appendix. Pandemic metaphor category.**
(DOCX)

**S2 Appendix. Topics of the pandemic metaphors.**
(DOCX)

## Acknowledgments

We are grateful to Dr. Graham Low formerly of the University of York for proofreading the manuscript. We thank three anonymous reviewers for offering insightful comments on the original version of the manuscript.

## Author Contributions

**Conceptualization:** Yu Deng, Wan Wan.

**Data curation:** Jixue Yang.

**Funding acquisition:** Yu Deng.

**Investigation:** Yu Deng, Jixue Yang.

**Methodology:** Yu Deng, Jixue Yang.

**Writing – original draft:** Yu Deng.

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
