## [Decision Letter · Decision Letter 0]

1 Dec 2021

PONE-D-21-35424Embodied Metaphor in Communication about Lived Experiences of the COVID-19 Pandemic in Wuhan, ChinaPLOS ONE

Dear Dr. Deng,

Thank you for submitting your manuscript to PLOS ONE. After careful consideration, we feel that it has merit but does not fully meet PLOS ONE’s publication criteria as it currently stands. Therefore, we invite you to submit a revised version of the manuscript that addresses the points raised during the review process.

We look forward to receiving your revised manuscript.

Kind regards,

Sanjay Kumar Singh Patel, Ph.D.

Academic Editor

PLOS ONE

Journal Requirements:

(This study was supported by Humanities and Social Sciences Research Project of Chongqing Education Commission (21SKGH143) and Foundation of First-class Discipline of Foreign Languages & Literature, Chongqing (SISUWYJY202104).)

(No potential conflict of interest was reported by the authors.)

(This research was funded by the first author’s grant on COVID-19 and mental health)

 (This study was supported by the first author's grants: Humanities and Social Sciences Research Project of Chongqing Education Commission (21SKGH143) and Foundation of First-class Discipline of Foreign Languages & Literature, Chongqing (SISUWYJY202104).)

Additional Editor Comments (if provided):

Reviewers' comments:

Reviewer's Responses to Questions

**Comments to the Author**

1. Is the manuscript technically sound, and do the data support the conclusions?

Reviewer #1: Yes

Reviewer #2: Yes

Reviewer #3: Yes

2. Has the statistical analysis been performed appropriately and rigorously? 

Reviewer #1: Yes

Reviewer #2: Yes

Reviewer #3: Yes

3. Have the authors made all data underlying the findings in their manuscript fully available?

Reviewer #1: Yes

Reviewer #2: Yes

Reviewer #3: Yes

4. Is the manuscript presented in an intelligible fashion and written in standard English?

Reviewer #1: Yes

Reviewer #2: Yes

Reviewer #3: Yes

5. Review Comments to the Author

Reviewer #1: The manuscript by Deng et al. “Embodied Metaphor in Communication about Lived Experiences of the COVID-19 Pandemic in Wuhan, China” is an interesting study about citizens employed metaphors to communicate about their lived experiences of the COVID-19 pandemic. The examination of metaphors echoed the different kinds of emotional states and psychological conditions of the research participants, focusing on their mental imagery of COVID-19, extreme emotional experiences, and symbolic behaviors under the pandemic. Authors have concluded that bodily experiences of the pandemic, the environment, and the psychological factors combine to shape people’s metaphorical thinking processes. This study is noteworthy/interesting but the manuscript requires minor revision before its publication.

Comments

1. The English of manuscript can be polished (minor).

2. At least one additional Figure (illustration) may be provided as to highlight the summary or prospect of this study.

3. The abbreviations should be cross validated in the manuscript (First define them fully followed by abbreviation) and one paragraph can be added for abbreviations.

4. Authors should discuss about the limitation to their study.

Reviewer #2: In this paper entitled "Embodied Metaphor in Communication about Lived Experiences of the COVID-19 Pandemic in Wuhan, China", the authors investigate a group of 27 Wuhan citizens employed metaphors to communicate about the livid experience of covid-19 pandemic. The results show that multiple factors such as bodily experience, environment, and psychological factors shape people's thinking. The manuscript is good and well carried out. However, it requires revision to address minor comments.

Minor comments:

1) The study investigated a group of 27 Wuhan citizens. Could the authors explain how his results are significant with a small sample size in the manuscript?

2) Introduction, the importance of this study may be more specifically highlighted.

3) The author may provide a paragraph regarding challenges or prospects of study in the manuscript.

4) The authors may additionally provide one Figure:

4a) The participant information and interview process may be represented in tabular and flowchart form, respectively.

4b) Results may be represented in graphical form (Appendix A & B).

Reviewer #3: It is a well-written report of extensive research work on the Covid-19 pendamic, with very well planned experimental work and parameters chosen. The overall quality of manuscript is good but still there are many grammatically and spelling mistakes in this article which must be taken care of in the revised version so that it could publish.

---

## [Author Response · Author response to Decision Letter 0]

11 Dec 2021

Dear Dr. Patel and Reviewers,

We appreciate the interest that you and the reviewers have taken in our manuscript entitled “Embodied Metaphor in Communication about Lived Experiences of the COVID-19 Pandemic in Wuhan, China”. The comments are very insightful and valuable for improving our manuscript. We have made revisions in accordance with the reviewers’ enlightening suggestions. Revised and rewritten portions are marked in dark blue in the revised manuscript. We have also enclosed the point-by-point response to the reviewer’s comments (see the attached file).

The data availability statement and funding are updated accordingly:

Data Availability Statement: The minimal dataset is available within the paper. Additional data is available in figshare: https://doi.org/10.6084/m9.figshare.17152463

Funding: This study was supported by Humanities and Social Sciences Research Project of Chongqing Education Commission (21SKGH143) and Foundation of First-class Discipline of Foreign Languages & Literature, Chongqing (SISUWYJY202104). There was no additional external funding received for this study.

Competing interests: The authors have declared that no competing interests exist.

Thank you for your interest and efforts and we are looking forward to hearing from you in due time regarding our manuscript, and we are ready to respond to any further questions and comments you may have.

Thanks and best wishes,

The authors

Response to Reviewers 

Reviewer #1

The manuscript by Deng et al. “Embodied Metaphor in Communication about Lived Experiences of the COVID-19 Pandemic in Wuhan, China” is an interesting study about citizens employed metaphors to communicate about their lived experiences of the COVID-19 pandemic. The examination of metaphors echoed the different kinds of emotional states and psychological conditions of the research participants, focusing on their mental imagery of COVID-19, extreme emotional experiences, and symbolic behaviors under the pandemic. Authors have concluded that bodily experiences of the pandemic, the environment, and the psychological factors combine to shape people’s metaphorical thinking processes. This study is noteworthy/interesting but the manuscript requires minor revision before its publication.

Comments：

1. The English of manuscript can be polished (minor).

Reply: We have the revised manuscript polished by a native speaker from UK (i.e. a professor in linguistics). The revised and added contents are marked in dark blue.

2. At least one additional Figure (illustration) may be provided as to highlight the summary or prospect of this study.

Reply: We have added one Figure at the end of the revised manuscript to summarize the main points of this study on page 15.

3. The abbreviations should be cross validated in the manuscript (First define them fully followed by abbreviation) and one paragraph can be added for abbreviations.

Reply: We have cross validated all the abbreviations and added one paragraph at the end of the manuscript, as below on page 15:

“Abbreviations: COVID-19 refers to Corona Virus Disease; SARS represents Severe Acute Respiratory Syndrome; AIDS and HIV are the abbreviated forms of Acquired Immunodeficiency Syndrome and Human Immunodeficiency Virus, respectively; MIP and MIPVU denote metaphor identification procedure and metaphor identification procedure of VU University Amsterdam, respectively; 4E approach to embodied cognition means embodied, embedded, enacted, extended approaches.”(See page 15)

4. Authors should discuss about the limitation to their study.

Reply: We have provided two paragraphs regarding limitations and future research following the Discussion section. The added paragraphs can be seen on page 14, as copied below:

Study limitations and future research

This study has some limitations. First, we examined metaphors in the framing of the COVID-19 pandemic under a collective emotion [26]. It assumed that participants from Wuhan would have the same directions or tendencies as regards emotions or communicating emotions towards the COVID-19 experiences. Although participants were recruited from the same city, the metaphor informants were of different types. Each of their perspectives could be different with respect to their COVID-19 experiences. A second limitation has to do with the small sample size. This study involved a qualitative analysis of pandemic metaphors in a convenience sample of 27 participants from Wuhan during COVID-19. The results may not be generalizable to a wider population. Furthermore, the lack of face-to-face interviews might have prevented participants from freely expressing their possible frustrations during the pandemic [35]. 

 In the light of the potential challenges, future research could employ large samples of video or face-to-face interviews to complement the present research. Furthermore, quantitative models can be used to distinguish different metaphorical conceptualizations of COVID-19 among the different types of participants. Individual background factors such as gender, age, and quarantine time can be powerful predictors in exploring the framing variation of metaphor in communication about the lived experiences of the pandemic.” (See pages 14)

Reviewer #2

In this paper entitled "Embodied Metaphor in Communication about Lived Experiences of the COVID-19 Pandemic in Wuhan, China", the authors investigate a group of 27 Wuhan citizens employed metaphors to communicate about the livid experience of covid-19 pandemic. The results show that multiple factors such as bodily experience, environment, and psychological factors shape people's thinking. The manuscript is good and well carried out. However, it requires revision to address minor comments.

Minor comments:

1) The study investigated a group of 27 Wuhan citizens. Could the authors explain how his results are significant with a small sample size in the manuscript?

Reply: Thanks for this constructive comment. We have added the challenges of this study regarding the small sample size in the newly added part “Study limitations and future research” on page 14, as copied below:

 “…A second limitation has to do with the small sample size. This study involved a qualitative analysis of pandemic metaphors in a convenience sample of 27 participants from Wuhan during COVID-19. The results may not be generalizable to a wider population. …

 In the light of the potential challenges, future research could employ large samples of video or face-to-face interviews to complement the present research. “

 Looking at the literature of the qualitative study paradigm, the small sample can still yield significant findings. For example:

 Gibbs & Franks [3] analyzed metaphors in six women’s narratives of cancer experiences. The results showed that metaphors predominately originated from ordinary embodied experiences relating to the healthy body, such that patients concentrated on certain metaphorical ways of viewing their illnesses unique to their individual experiences [15]. (See page 1, the introduction of our paper)

 Gibbs R, Franks H. Embodied metaphor in women’s narratives about their experiences with cancer. Health Communication. 2002; 14(2): 139–65. https://doi.org/10.1207/S15327027HC1402_1

 Another example is that in a recent PLOS ONE paper, Falvo et al. (2021) conducted a qualitative study in a convenience sample of merely 19 participants from Switzerland during the first COVID-19 lockdown and reported some significant findings regarding their lived experiences during the pandemic.

 Falvo I, Zufferey MC, Albanese E, Fadda M. Lived experiences of older adults during the first COVID-19 lockdown: A qualitative study. PLoS ONE. 2021 June; 16(6): e0252101. https://doi.org/10.1371/journal.pone.0252101

In the future research, we will definitely draw on large samples, continuing to validate the findings of the present study.

2) Introduction, the importance of this study may be more specifically highlighted.

Reply: We have added a paragraph highlighting the importance of this study at the end of “Introduction” on page 3 of the revised manuscript, as copied below:

 “Essentially, we argue that metaphor can provide functional and concrete cognitive patterns in articulating and constructing participants’ subjective lived experiences, offering in-depth insight into their emotional states, identifying personal mental health issues and ultimately promoting positive changes. The hope is that the present paper will serve to shed light on the ways that metaphor can be employed by mental health professionals when evaluating and measuring people’s mental health before implementing effective mental health support and promoting accurate health awareness in the fight against the pandemic crisis. ” (See page 3)

3) The author may provide a paragraph regarding challenges or prospects of study in the manuscript.

Reply: Thanks for this constructive comment. We have added two paragraphs regarding limitations and future research at the end of the “Discussion” section. See the added paragraphs on page 14, as copied below:

Study limitations and future research

“This study has some limitations. First, we examined metaphors in the framing of the COVID-19 pandemic under a collective emotion [26]. It assumed that participants from Wuhan would have the same directions or tendencies as regards emotions or communicating emotions towards the COVID-19 experiences. Although participants were recruited from the same city, the metaphor informants were of different types. Each of their perspectives could be different with respect to their COVID-19 experiences. A second limitation has to do with the small sample size. This study involved a qualitative analysis of pandemic metaphors in a convenience sample of 27 participants from Wuhan during COVID-19. The results may not be generalizable to a wider population. Furthermore, the lack of face-to-face interviews might have prevented participants from freely expressing their possible frustrations during the pandemic [35]. 

 In the light of the potential challenges, future research could employ large samples of video or face-to-face interviews to complement the present research. Furthermore, quantitative models can be used to distinguish different metaphorical conceptualizations of COVID-19 among the different types of participants. Individual background factors such as gender, age, and quarantine time can be powerful predictors in exploring the framing variation of metaphor in communication about the lived experiences of the pandemic.” (See pages 14)

4) The authors may additionally provide one Figure:

4a) The participant information and interview process may be represented in tabular and flowchart form, respectively.

Reply: Thanks for the kind reminder. We have added a tabular (Table 1) about the participant information in the Participants section and a flowchart form (Figure 1) in the section of interview procedures. See Table 1 on page 4 and Figure 1 on page 5, respectively.

4b) Results may be represented in graphical form (Appendix Table A & B).

Reply：We have used a graphical form to represent the frequency distribution of the 49 pandemic metaphor categories. This figure corresponds to Appendix Table A. See Figure 4 on page 7

 As regards Appendix Table B, there are 70 topics for the pandemic metaphors; it appears to be difficult to represent them in a single graphical form. So we reserved the word cloud (Figure 3) to show the distribution of the 70 topics.

Reviewer #3

It is a well-written report of extensive research work on the Covid-19 pandemic, with very well planned experimental work and parameters chosen. The overall quality of manuscript is good but still there are many grammatically and spelling mistakes in this article which must be taken care of in the revised version so that it could publish.

Reply: Thanks for your great efforts and encouraging comments. The revised manuscript was polished by a native speaker from UK (i.e. a professor in linguistics). The revised and added contents are marked in dark blue.

---

## [Editor Report · Decision Letter 1]

15 Dec 2021

Embodied Metaphor in Communication about Lived Experiences of the COVID-19 Pandemic in Wuhan, China

PONE-D-21-35424R1

Dear Dr. Deng,

We’re pleased to inform you that your manuscript has been judged scientifically suitable for publication and will be formally accepted for publication once it meets all outstanding technical requirements.

Kind regards,

Sanjay Kumar Singh Patel, Ph.D.

Academic Editor

PLOS ONE

---

## [Editor Report · Acceptance letter]

19 Dec 2021

PONE-D-21-35424R1 

Embodied Metaphor in Communication about Lived Experiences of the COVID-19 Pandemic in Wuhan, China 

Dear Dr. Deng:

I'm pleased to inform you that your manuscript has been deemed suitable for publication in PLOS ONE. Congratulations! Your manuscript is now with our production department. 

Kind regards, 

on behalf of

Dr. Sanjay Kumar Singh Patel 

Academic Editor

PLOS ONE